# Prevalence of vitamin D and calcium deficiencies and their health impacts on women of childbearing age: a protocol for systematic review and meta-analysis

Erika Aparecida Silveira ,[1,2] Letícia de Almeida Nogueira e Moura,[1] Maria Clara Rezende Castro,[1] Gilberto Kac ,[3] Priscilla Rayanne e Silva Noll ,[4,5] Cesar de Oliveira ,[2] Matias Noll [6,7]

**Correspondence to**
PhD Erika Aparecida Silveira; erikasil@terra.com.br

## ABSTRACT

**Introduction** No systematic reviews has synthesised data on the available evidence to determine the prevalence of calcium and vitamin D deficiencies as a public health problem globally. Therefore, this study presents a protocol for conducting a review and meta-analysis to estimate the prevalence of calcium and vitamin D serum deficiencies in women of childbearing age and stratify these data by age group, urban and rural area, world region and pregnant/non-pregnant women whenever possible.

**Methods and analysis** The systematic review protocol involves conducting a literature search in the following databases: PubMed, LILACS, Embase, Scopus and Web of Science. The selected articles will be checked thoroughly, including the references to include grey literature. Cross-sectional studies and baseline data from cohort studies or clinical and community trials conducted with women of childbearing age with representative probabilistic sampling will be included. Two independent researchers will be responsible for article selection and data extraction, and discrepancies, if any, will be dealt with by a third reviewer. Methodological quality and risk of bias will be analysed using the Grading of Recommendations, Assessment, Development and Evaluations and Joanna Briggs Institute's checklist, respectively. The heterogeneity of the estimates between studies will also be evaluated. Dissemination of the key findings from the systematic review will help identify priorities for action, establish dietary guidelines, develop health-related public policies and reduce and combat micronutrient deficiencies among women of childbearing age and their children.

**Ethics and dissemination** Formal ethical approval is not required, and findings will be published in a peer-reviewed journal.

**PROSPERO registration number** CRD42020207850.

## Strengths and limitations of this study

⇒ This systematic review intends to estimate the prevalence of calcium and vitamin D serum deficiencies in women of childbearing age and stratify these data by age group, urban and rural area, world region and pregnant/non-pregnant women whenever possible.
⇒ The following five databases will be included to identify scientific articles: PubMed, LILACS, Embase, Scopus and Web of Science.
⇒ Meta-analysis will be evaluated using fixed-effects or random-effects methods according to the studies' homogeneity whenever possible.
⇒ Potential heterogeneity of measurements and assessments of micronutrient deficiency in research may be a limitation.

## INTRODUCTION

Nutritional deficiencies have multiple aetiologies related to factors such as nutrient loss, malabsorption, inadequate intake and even ignorance of the importance of nutritional care during pregnancy.[1] Women of childbearing age are a part of the highly vulnerable population and often experience these deficiencies, especially during pregnancy.[2–5] Besides, maternal nutritional status influences the pregnancy outcomes and child's health and, as it is necessary to supply the needs of both the mother and the fetus.[3–5]

Vitamin D deficiency is considered a significant public health problem and affects approximately one billion people worldwide.[6 7] However, there are significant data gaps in measuring this problem magnitude in many countries.[8] The estimated prevalence of vitamin D deficiency (20 ng/mL) was around 37% in Canada and 40% in Europe.[9] A systematic review with a meta-analysis of observational studies found a significant association between low vitamin D levels and all-cause mortality.[10] Regarding the effectiveness of vitamin D on specific diseases on this population, evidences still are not enough, and they should only refer to meta-analyses of randomised trials.[10–14] Specifically for pregnant women, systematic reviews with meta-analysis of randomised clinical trials found

that vitamin D supplementation can effectively prevent pre-eclampsia[11] and reduce maternal insulin resistance.[12] Furthermore, it is associated with high circulating levels of 25(OH)D, birth weight and length,[13] and fetal or neonatal mortality[14]

Vitamin D levels interfere with calcium metabolism.[15] Calcium is a micronutrient that plays an essential role in muscle contraction and regulates water balance in cells.[16] Hypocalcaemia symptoms, characterised by low calcium levels, are variable, ranging from mild disorders such as perioral paraesthesia to severe life-threatening ones such as cardiac arrhythmias.[17] In pregnant women, this imbalance is characterised by a lower serum concentration and a higher concentration in the cells, thereby causing increased blood pressure in women with pre-eclampsia.[18 19]

Previous studies have evaluated the prevalence of vitamin D and calcium deficiencies in women of childbearing age.[20–22] However, at present, there is no systematic review that synthesises the state of art of this relevant public health problem nationally and globally. Therefore, this study aimed to estimate the prevalence of calcium and vitamin D serum deficiencies in women of childbearing age. In addition, the data will be stratified by age group, urban and rural areas, geographical location and pregnant/non-pregnant women through a systematic review followed by a meta-analysis.

## METHODS AND ANALYSIS

This systematic review and meta-analysis will be written according to the Preferred Reporting Items for Systematic Reviews and Meta-Analysis Protocols. The protocol for this systematic review was registered on the PROSPERO platform. This review follows the Population, Intervention, Comparison and Outcome (PICO) structure, with the P- population being women of childbearing age, the I- intervention being no intervention on this study, the C- comparison being conducted between subgroups in childbearing age as pregnant and not pregnant and the O-outcome being calcium and vitamin D deficiencies. Therefore, our systematic review will be conducted based on the following research question: 'What is the prevalence of calcium and vitamin D serum deficiencies in women of childbearing age?'.

## Search strategy and databases

A systematic review of the literature will be conducted to identify and collect bibliographic data in the following databases: PubMed (National Library of Medicine, USA), LILACS (Latin American and Caribbean Health Sciences Literature), Embase, Scopus and Web of Science databases (figure 1). The databases will be searched in December 2021 to identify the potential articles to be included in the systematic review. The search strategy will be complemented by screening the reference list of the included studies and relevant systematic reviews on the following months.

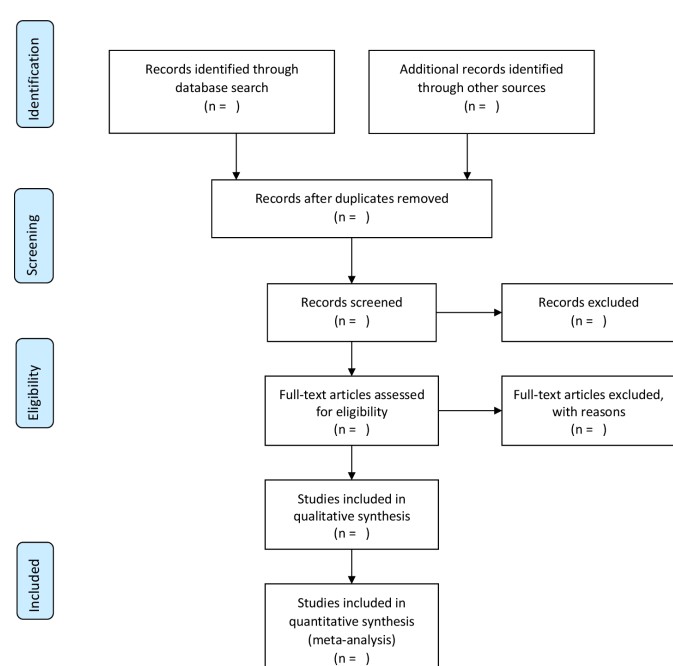

**Figure 1** Preferred Reporting Items for Systematic Reviews and Meta-Analyses flow diagram.

Different search strategies will include medical subject heading terms and relevant keywords related to vitamin D deficiency and specific terms related to calcium deficiency to ensure that all articles of interest are identified using all the descriptors and Boolean operators of this systematic review protocol (online supplemental table).

## Eligibility criteria
### Inclusion criteria
► Studies that provide data on the prevalence of calcium and/or vitamin D serum deficiency in women of childbearing age (15–49 years old or menarche and menopause).
► Studies with representative population-based samples in hospitals, health centres or outpatient clinics.
► Prevalence data on women of different age groups, such as adolescents, pregnant women, lactating women and premenopausal adult women.
► Studies with cross-sectional design and data from longitudinal studies (cohort studies) or intervention studies such as clinical trials or community trials providing they have prevalence information in a specific point of time. Articles in English, Portuguese and Spanish languages were selected.

### Exclusion criteria
► Opinion articles, comments or editorials.
► Duplicates: the most comprehensive one that reports the largest sample size will be considered for studies published as more than one article.
► Studies with primary data not accessible even after request.
► Case–control studies, narrative reviews and case series.

► Studies conducted among athlete women of any sports modality.
► Studies conducted among women with specific diseases:
- Autoimmune diseases: lupus, psoriasis, thyroiditis, rheumatoid arthritis and multiple sclerosis.
- Eating disorders: anorexia and bulimia.
- Haematological diseases: thalassaemia and sickle cell disease.
- Respiratory diseases: Chronic obstructive pulmonary disease (COPD), asthma, pneumonia, respiratory infections and tuberculosis.
- Chronic diseases such as: heart failure, kidney failure, liver diseases, chronic kidney disease, cardiac diseases, nephrotic syndrome, AIDS, inflammatory bowel disease, hypothyroidism, sepsis and cancer.
- Genetic diseases and syndromes: mutation in the vitamin D receptor, cystic fibrosis and Prader-Willi syndrome.
- Neurological or psychiatric illnesses: epilepsy or use of antiepileptic medication, Attention-deficiency hiperactivity disorder (ADHD) and schizophrenia.
- Trauma, post-surgical: burns, post-bariatric, in the treatment of recent fractures or orthopaedic/osteoarticular diseases.
- Patients in intensive care, urgency and emergency and palliative care.
- Women with hypo and hyperparathyroidism.
- Small number of participants (<50).
- Studies conducted among indigenous women.

## Reviewers' training
The authors responsible for assessing article eligibility will be trained on using the inclusion and exclusion criteria. Thus, an eligibility test will be conducted on 50 titles and abstracts before codifying the articles. They will also receive training on instruments to assess the risk of bias used on five articles that have not been included. Rayyan and Mendeley software will be used for the selection steps.

## Review process
After completing the search strategy, the identified articles will be gathered and imported into Mendeley software. Duplicate articles will be deleted. The articles will be selected by two independent reviewers (LdANeM and MCRC). The titles will be read in the first stage, and the abstracts in the second stage. Finally, in the third stage, the entire article will be read. Disagreements between reviewers will be assessed by a third reviewer (EAS). Eligibility will be determined according to the inclusion and exclusion criteria. The percentage of agreement and the Kappa coefficient will be calculated to assess reliability between reviewers' assessments. At the end of the process, the selected articles will be included in the systematic review.

## Data extraction and risk of bias assessment
Data extraction will be performed using a spreadsheet that considers the following aspects: author/year, type of study, place of study, age group, type of sample/sample size, place of residence (urban/rural), continent/world region (Asia, Europe, Africa, Latin America, North America and Oceania), pregnancy status (yes/no), lactating (yes/no), micronutrients analysed, the technique used, cut-off points and results, that is, prevalence/impact of calcium or vitamin D deficiencies (table 1).

In this study, we will use the Joanna Briggs Institute's development tool of a critical appraisal tool for use in systematic reviews addressing questions of prevalence.[23] This tool has excellent methodological rigour, handling essential items related to the methodological quality of the studies.[24] To ensure the quality of the study, the tool consists of 10 questions that assess criteria such as representativeness and sample size, recruitment of participants and data analysis.[23]

The Grading of Recommendations, Assessment, Development and Evaluations will be used to assess the strength of evidence.[25] The quality of evidence will be classified into four grades: high, moderate, low or very low.[25]

The data will be extracted and evaluated by two independent reviewers (LdANeM and MCRC). Disagreements will be resolved by a third reviewer (MN). To obtain relevant data not described in the manuscript, a researcher (MCRC) will contact the article's authors. The authors of the included studies' potential conflicts of interest and ethical information will also be described.

## Patient and public involvement
No patient involved.

## Statistical analysis
Meta-analysis will be performed using random-effects methods. $\chi^2$ test with a significance level ($p<0.05$) will be used to assess heterogeneity, and the $I^2$ statistic will be used to determine the magnitude of inconsistency. $I^2$ results above 75% will indicate a high heterogeneity, those between 25% and 75% will characterise moderate heterogeneity and those below 25% will indicate low heterogeneity.[26–28]

According to the available literature, sensitivity and estimation of the prevalence of vitamin D and calcium deficiencies will be analysed. The data will be stratified according to region, age and urban/rural area subgroups. If applicable, meta-regression will be performed to explain the studies' heterogeneity considering $p<0.05$ for each stratification subgroup. We will estimate the pooled prevalences by obtaining the following information from each study: the number of women with calcium and vitamin D deficiencies and sample size total. Analyses will be

conducted using R through the package Meta. The results will be presented as prevalence with 95% CI.

The primary outcome of the present study is the prevalence of serum calcium and vitamin D deficiencies in women of childbearing age.

The prevalence of circulating 25(OH)D concentrations below certain thresholds, such as the percentage of values <30 nmol/L or <50 nmol/L, will be presented. Selected studies will be grouped in descriptive tables according to their cut-offs. This way, we will highlight and discuss the current controversies and the need for a standard classification of vitamin D status. This will be one of the scientific contributions of our systematic review.

## DISCUSSION

Nutritional deficiencies affect pregnant and non-pregnant women, especially those living in poorer regions like the Brazilian northeast.[29–33] Thus, such nutritional deficiencies need to be reviewed nationally and globally as public health problems.[34 35] Therefore, it is essential to estimate the extension of this problem. To the best of our knowledge, there are no systematic reviews and meta-analyses that synthesise data on the available evidence to assess the extent of the prevalence of calcium and vitamin D deficiencies in women of childbearing age.

Obstacles in implementing policies and public health programmes to combat nutritional deficiencies can be exacerbated by inequalities in health service provision and access.[36] Therefore, it is essential to gather, organise and systematise the available data on the prevalence of the main vitamin and mineral deficiencies to develop intervention plans, list action priorities and improve existing public health policies for women of childbearing age.

This systematic review protocol is important to improve knowledge on global and Brazilian realities on the prevalence of vitamin D and calcium deficiency in this population group. This study is expected to contribute to suggestions for priority action strategies and the development of public health policies to minimise the problem and prevent the occurrence of new cases and assist in establishing intervention protocols to reduce the main deficiencies according to the priorities indicated in the final report. However, some limitations of this research need to be highlighted. First, the potential heterogeneity of measurements and assessments of micronutrient deficiency in research. Second, studies may have used different methods, making it difficult to conduct a meta-analysis.

Women of childbearing age, especially pregnant women, are a part of a highly vulnerable group that develops nutritional deficiencies, and the maternal nutritional status influences gestational outcomes and the child's health. Considering the above, assessing the prevalence of vitamin D and calcium deficiencies

**Table 1** Data extraction worksheet

| Author/ year | Study type | Study location | Age group | Sample type/ sample size | Place of residence (rural/urban) | Continent/ world region | Pregnant women (yes/no) | Lactating women (yes/no) | Micronutrients analysed | | Technique used | Cut-off points | Prevalence | | Notes |
|---|---|---|---|---|---|---|---|---|---|---|---|---|---|---|---|
| | | | | | | | | | Calcium | Vitamin D | | | Calcium | Vitamin D | |

could help identify knowledge gaps on the magnitude of these deficiencies, warranting further studies. From this perspective, disseminating the results of this systematic review protocol will be helpful and vital to encourage research and identifying priorities, establishing dietary guidelines, and reducing and combating micronutrient deficiencies in this population.

## Author affiliations
[1]Health Science Graduate Program, Medicine Faculty, Federal University of Goiás, Goiânia, Brazil
[2]Department of Epidemiology and Public Health, University College London, London, UK
[3]Federal University of Rio de Janeiro, Rio de Janeiro, Brazil
[4]Department of Obstetrics and Gynecology, University of São Paulo, São Paulo, Brazil
[5]Instituto Federal Goiano, Ceres, Goiás, Brazil
[6]University of Southern Denmark, Odense, Denmark
[7]Insituto Federal Goiano, Ceres, Goiás, Brazil

**Acknowledgements** The authors thank CNPq and IF Goiano for providing partial support.

**Contributors** Conceptualisation, methodology and formal analysis: EAS, LdANeM, MCRC, CdO and MN. Investigation, writing—original draft preparation, writing—review and editing and visualisation: EAS, LdANeM, MCRC, GK, PN, CdO and MN. Resources and funding acquisition: EAS and CdO. Supervision: EAS. All authors have read and agreed to the published version of the manuscript.

**Funding** This study received support from the Conselho Nacional de Desenvolvimento Científico e Tecnológico (MS-SCTIE-Decit/CNPq N° 442932/2019–7). CdO is supported by the Economic and Social Research Council (grant number: ES/T008822/1).

**Competing interests** None declared.

**Patient and public involvement** Patients and/or the public were not involved in the design, or conduct, or reporting, or dissemination plans of this research.

**Patient consent for publication** Not applicable.

**Provenance and peer review** Not commissioned; externally peer reviewed.

## ORCID iDs
Erika Aparecida Silveira http://orcid.org/0000-0002-8839-4520
Gilberto Kac http://orcid.org/0000-0001-8603-9077
Priscilla Rayanne e Silva Noll http://orcid.org/0000-0003-3715-1956
Cesar de Oliveira http://orcid.org/0000-0001-8603-9077
Matias Noll http://orcid.org/0000-0002-1482-0718

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
