## [Reviewer comments · BMJ Open]

ARTICLE DETAILS

TITLE (PROVISIONAL)	Prevalence of vitamin D and calcium deficiencies and their health impacts on women of childbearing age: a protocol for systematic review and meta-analysis
AUTHORS	Silveira, Erika; Moura, Letícia; Castro, Maria; Kac, Gilberto; Noll, Priscilla; de Oliveira, Cesar; Noll, Matias

VERSION 1 – REVIEW

REVIEWER	Armin Zittermann Ruhr University Bochum
REVIEW RETURNED	21-Jul-2021

GENERAL COMMENTS	The present study presents a protocol for conducting a review and meta-analysis to estimate the prevalence of calcium and vitamin D deficiencies in women of childbearing age and stratify these data by age group, urban and rural area, world region, and pregnant/non-pregnant women. • It is a huge task to evaluate all representative data in women of childbearing age globally. Moreover, this reviewer is not sure, whether all data of representative dietary surveys are published in the mentioned databases, since sometimes surveys are published in local language only.• The definition of vitamin D deficiency according to circulating 25-hydroxyvitamin D is discussed controversially in the scientific community. Note that the former North American Institute of Medicine (now: National Academy of Medicine) considers only 25-hydroxyvitamin D concentrations below 12 ng/ml and not concentrations below 20 ng/ml, as mentioned in the manuscript, as deficient. Thus, classification of vitamin D status according to circulating 25-hydroxyvitamin D is crucial. This concern also applies to dietary calcium intake, since recommendations vary widely globally. Therefore, cut-offs used for their review will have a profound impact on the results of the meta-analysis.• Highlights: The wording 'first systemic review' should be omitted.• Introduction: They state that vitamin D deficiency (25-hydroxyvitamin D < 20 ng/ml?) has been associated with adverse events such as an increased risk of cardiovascular disease, infection, cancer, bone loss, fractures, and mortality. However, associations do not prove causality. The causal relationships of vitamin D with these diseases are less clear and they should only refer to meta-analyses of randomized trials, which are available for all mentioned diseases.
---

	 • Vitamin D and dietary calcium can replace each other with respect to serum calcium homeostasis and the risk of hypocalcemia. How do they want to handle results of high dietary calcium intake and low vitamin D, and vice versa.
--	---

REVIEWER	Cintia Curioni Universidade do Estado do Rio de Janeiro
REVIEW RETURNED	30-Aug-2021

GENERAL COMMENTS	The protocol is well written and the topic addressed is very relevant, since women of childbearing age are usually neglected in studies. My concern is related to the dates of study. In the protocol registered on the PROSPERO platform, the anticipated completion date registered is 01 August 2021. The authors did not include any dates of the study. If the study is already done, publishing the protocol does not make sense to me. Specific Comments:  1. The addressed question should be included in the methods of the protocol. 2. The “Downs and Black tool” was used to analyze the quality and risk of bias, adapted for observational studies. The cited reference (29) did not show the validity of the adapted tool. The risk of bias was not even assessed in this study. One option could be the JBI critical appraisal tool, which has been formally evaluated and is increasingly used across these types of reviews (Munn Z, Moola S, Riitano D, Lisy K. The development of a critical appraisal tool for use in systematic reviews addressing questions of prevalence. Int J Health Policy Manag. 2014;3(3):123–8). 3. Statistical analysis: more details could be provided. It is important to mention the choice of methods to transform estimates for proportional meta-analysis.
---

VERSION 1 – AUTHOR RESPONSE

Reviewer #1:

The present study presents a protocol for conducting a review and meta-analysis to estimate the prevalence of calcium and vitamin D deficiencies in women of childbearing age and stratify these data by age group, urban and rural area, world region, and pregnant/non-pregnant women. It is a huge task to evaluate all representative data in women of childbearing age globally. Moreover, this reviewer is not sure, whether all data of representative dietary surveys are published in the mentioned databases, since sometimes surveys are published in local language only.

Thank you for raising this important point. We appreciate the reviewer's concern. The authors are aware of the limitations with regards to how representative dietary surveys are, how food intake information is collected as well as the bias around dietary consumption/intake. Therefore, to address such limitations, our systematic review will focus on serum levels of these micronutrients. We will also use all data published in English, Spanish and Portuguese. We have revised the manuscript to make this information clearer.

- The definition of vitamin D deficiency according to circulating 25-hydroxyvitamin D is discussed controversially in the scientific community. Note that the former North American Institute of Medicine (now: National Academy of Medicine) considers only 25-hydroxyvitamin D concentrations below 12 ng/ml and not concentrations below 20 ng/ml, as mentioned in the manuscript, as deficient. Thus, classification of vitamin D status according to circulating 25-hydroxyvitamin D is crucial. This concern

also applies to dietary calcium intake, since recommendations vary widely globally. Therefore, cut-offs used for their review will have a profound impact on the results of the meta-analysis.

We agree with the points raised by the reviewer. The authors would like to clarify that we will not consider a specific cutoff point as an eligibility criterion, based on the existence of different classifications in the literature. Because the cut-offs to be used in our review will have a profound impact on the results of this study, this information will be in our data extraction Table. In this study, we will not work with dietary intake analysis, but with serum levels.

- Highlights: The wording ‘first systemic review’ should be omitted.

Thank you for your suggestion. We have removed the mentioned words from the Highlights as requested.

- Introduction: They state that vitamin D deficiency (25-hydroxyvitamin D < 20 ng/ml?) has been associated with adverse events such as an increased risk of cardiovascular disease, infection, cancer, bone loss, fractures, and mortality. However, associations do not prove causality. The causal relationships of vitamin D with these diseases are less clear and they should only refer to meta-analyses of randomized trials, which are available for all mentioned diseases.

Thank you for pointing this out. We have included information on these causal relationships, as suggested, on the second paragraph as follows:

“A significant association between low vitamin D levels and all-cause mortality was found in a systematic review with meta-analysis of observational studies [10]. The causal relationships of vitamin D with diseases are less clear and they should only refer to meta-analyses of randomized trials [10–14]. Specifically for pregnant women, systematic reviews with meta-analysis of randomized clinical trials found that vitamin D supplementation can effectively prevent pre-eclampsia [11] and reduce maternal insulin resistance [12]. Furthermore, it is associated with high circulating levels of 25(OH)D, birth weight and length [13], and fetal or neonatal mortality [14].”

We have added the extra references below from systematic reviews with meta-analysis of randomized clinical trials to the revised Introduction:

- Garland CF, Kim JJ, Mohr SB, *et al.* Meta-analysis of All-Cause Mortality According to Serum 25-Hydroxyvitamin D. *American Journal of Public Health* 2014;**104**:e43–50. doi:10.2105/AJPH.2014.302034
- Fogacci S, Fogacci F, Banach M, *et al.* Vitamin D supplementation and incident preeclampsia: A systematic review and meta-analysis of randomized clinical trials. *Clinical Nutrition* 2020;**39**:1742–52. doi:10.1016/j.clnu.2019.08.015
- Gallo S, McDermid JM, Al-Nimr RI, *et al.* Vitamin D Supplementation during Pregnancy: An Evidence Analysis Center Systematic Review and Meta-Analysis. *Journal of the Academy of Nutrition and Dietetics* 2020;**120**:898-924.e4. doi:10.1016/j.jand.2019.07.002
- Pérez-López FR, Pasupuleti V, Mezones-Holguin E, *et al.* Effect of vitamin D supplementation during pregnancy on maternal and neonatal outcomes: a systematic review and meta-analysis of randomized controlled trials. *Fertility and Sterility* 2015;**103**:1278-1288.e4. doi:10.1016/j.fertnstert.2015.02.019
- Bi WG, Nuyt AM, Weiler H, *et al.* Association Between Vitamin D Supplementation During Pregnancy and Offspring Growth, Morbidity, and Mortality. *JAMA Pediatrics* 2018;**172**:635. doi:10.1001/jamapediatrics.2018.0302

- Vitamin D and dietary calcium can replace each other with respect to serum calcium homeostasis and the risk of hypocalcemia. How do they want to handle results of high dietary calcium intake and low vitamin D, and vice versa.

Thank you. Our systematic review will focus on serum data due to the limitations and bias of dietary information. For example, with regards to the deficiency prevalence we have decided to use only serum information of the micronutrients in question.

Reviewer #2:

The protocol is well written and the topic addressed is very relevant, since women of childbearing age are usually neglected in studies.

Thank you for your kind words. Much appreciated.

My concern is related to the dates of study. In the protocol registered on the PROSPERO platform, the anticipated completion date registered is 01 August 2021. The authors did not include any dates of the study. If the study is already done, publishing the protocol does not make sense to me.

Thank you. The systematic literature review is delayed due to the impact of COVID-19 pandemic on our schedule. We apologize for failing to previously update the schedule in the Prospero platform. The timeline in Prospero has now been updated. We have added the following statement in the revised Methods section, search strategy and databases topic as follows: In December 2021, a systematic review of the literature will be conducted....”

Specific Comments:

1. The addressed question should be included in the methods of the protocol.

Thank you. The research question to be addressed by our systematic review has been included at the beginning of the revised Methods section as follows:

“Therefore, our systematic review will be conducted based on the following research question: ‘What is the prevalence of calcium and vitamin D serum deficiencies in women of childbearing age?’”

2. The “Downs and Black tool” was used to analyze the quality and risk of bias, adapted for observational studies. The cited reference (29) did not show the validity of the adapted tool. The risk of bias was not even assessed in this study. One option could be the JBI critical appraisal tool, which has been formally evaluated and is increasingly used across these types of reviews (Munn Z, Moola S, Riitano D, Lisy K. The development of a critical appraisal tool for use in systematic reviews addressing questions of prevalence. *Int J Health Policy Manag.* 2014;3(3):123–8).

Thank you very much for your valuable suggestion. We have changed the instrument to assess the quality and risk of bias to the JBI critical appraisal tool as suggested. Please, see the revised text on page 8.

3. Statistical analysis: more details could be provided. It is important to mention the choice of methods to transform estimates for proportional meta-analysis.

Thank you for your suggestion. We have included more information on the choice of methods to transform estimates for proportional meta-analysis in the revised Methods section as suggested.

VERSION 2 – REVIEW

REVIEWER	Armin Zittermann Ruhr University Bochum
REVIEW RETURNED	02-Nov-2021

GENERAL COMMENTS	The manuscript has been improved, but some issues remain.  • They argue that they will not consider a specific cutoff point for classification of vitamin D deficiency. This is understandable given the ongoing debate about classification of vitamin D status based on circulating 25(OH)D. Nevertheless, it will at least be necessary to present the prevalence of circulating 25(OH)D concentrations below certain thresholds, such as the percentage of values <30 nmol/L or <50 nmol/L. Please clarify and specify. • They have decided to use only serum information for assessing calcium deficiency prevalence. Nevertheless, two issues remain. First, they need to establish reference values for appropriate serum calcium concentrations in women of childbearing age. Second, serum calcium concentrations outside the reference range are more indicative of endocrinologic disorders of calcium metabolism such as hypo- and hyperparathyroidism than of dietary calcium status. Therefore, the problem of assessing dietary calcium status in this population remains. • My comment 'The causal relationships of vitamin D with diseases are less clear and they should only refer to meta-analyses of randomized trials' cannot be transferred 1:1 into the manuscript. Please correct.
---

REVIEWER	Cintia Curioni Universidade do Estado do Rio de Janeiro
REVIEW RETURNED	28-Oct-2021

GENERAL COMMENTS	"In December 2021, a systematic review of the literature will be conducted to identify and collect bibliographic data in the following databases...". Since a systematic review is not carried out just in one month, it is necessary to change. It is important to state the date the databases were last searched. Regarding the Statistical analysis, the choice between a fixed-effect and a random-effects meta-analysis should never be made on the basis of a statistical test for heterogeneity: Under the fixed-effect model, we assume that there is one true effect size that underlies all the studies in the analysis, and under the random-effects model we allow that the true effect size might differ from study to study. Authors should consider, what model is more appropriate to analyze the data. I suppose the random effect is more appropriate in this situation since of high levels of heterogeneity between populations.
---

VERSION 2 – AUTHOR RESPONSE

Reviewer: 2

Dr. Cintia Curioni, Universidade do Estado do Rio de Janeiro

Comments to the Author:

“In December 2021, a systematic review of the literature will be conducted to identify and collect bibliographic data in the following databases...”. Since a systematic review is not carried out just in one month, it is necessary to change. It is important to state the date the databases were last searched.

Response: Thank you for your comment. We agree with the reviewer that it is impossible to carry out a systematic review in one month. The authors meant that the databases will be searched in December 2021 to identify the potential articles to be included in the systematic review. Apologies for the confusion. We have changed this information in the revised version of the manuscript to make it clearer.

Regarding the Statistical analysis, the choice between a fixed-effect and a random-effects meta-analysis should never be made on the basis of a statistical test for heterogeneity:

Under the fixed-effect model, we assume that there is one true effect size that underlies all the studies in the analysis, and under the random-effects model we allow that the true effect size might differ from study to study.

Authors should consider, what model is more appropriate to analyze the data. I suppose the random effect is more appropriate in this situation since of high levels of heterogeneity between populations.

Response: Thank you for raising this important point. We agree with the reviewer and have changed this information in the revised methods section. We have chosen random-effects.

Reviewer: 1

Dr. Armin Zittermann, Ruhr University Bochum

Comments to the Author:

The manuscript has been improved, but some issues remain.

- They argue that they will not consider a specific cutoff point for classification of vitamin D deficiency. This is understandable given the ongoing debate about classification of vitamin D status based on circulating 25(OH)D. Nevertheless, it will at least be necessary to present the prevalence of circulating 25(OH)D concentrations below certain thresholds, such as the percentage of values <30 nmol/L or <50 nmol/L. Please clarify and specify.

Response: Thank you for raising this important point. Firstly, the authors would like to highlight that the present manuscript is a systematic review protocol, and as such, we will consider what the original article had defined as vitamin D deficiency. However, to address the reviewer's point, we plan to present the prevalence of circulating 25(OH)D concentrations below certain thresholds, such as the percentage of values <30 nmol/L or <50 nmol/L as suggested by the reviewer. We will group the selected studies in descriptive Tables according to their cut-offs. This way, we will highlight and discuss the current controversies and the need for a standard classification of vitamin D status. This will be one of the scientific contributions of our systematic review.

- They have decided to use only serum information for assessing calcium deficiency prevalence. Nevertheless, two issues remain. First, they need to establish reference values for appropriate serum calcium concentrations in women of childbearing age. Second, serum calcium concentrations outside the reference range are more indicative of endocrinologic disorders of calcium metabolism such as hypo- and hyperparathyroidism than of dietary calcium status. Therefore, the problem of assessing dietary calcium status in this population remains.

Response: Thank you for your comment. As a systematic review protocol, we must consider what the original article had defined as calcium deficiency. Nevertheless, we will discuss all the cut-offs of the included studies and their limitations in the manuscripts to come. We will group the selected studies in descriptive Tables according to their cut-offs as published on original articles. We are interested in describing the occurrence of the problem i.e. calcium deficiency prevalence according to the inclusion and exclusion criteria detailed in the manuscript. We will add a new exclusion criterion for women of childbearing age with hypo and hyperparathyroidism. Our main objective is to describe the prevalence of serum calcium deficiency. It is our focus, and this is on the PROSPERO registry of this systematic review, and we cannot modify our objective.

- My comment 'The causal relationships of vitamin D with diseases are less clear and they should only refer to meta-analyses of randomized trials' cannot be transferred 1:1 into the manuscript. Please correct.

Response: Thank you for the comment. We have rewritten this paragraph in the revised Introduction as requested.